TOPICAL REVIEW

# The repair capacity spectrum of human skeletal muscle injury from sports to surgical trauma settings

Grith Højfeldt[1,2] , Christian Hoegsbjerg[1,2], Arvind G. von Keudell[3,4] and Abigail L. Mackey[1,2]

[1]*Institute of Sports Medicine Copenhagen, Department of Orthopaedic Surgery, Copenhagen University Hospital - Bispebjerg and Frederiksberg, Copenhagen, Denmark*

[2]*Department of Clinical Medicine, University of Copenhagen, Copenhagen, Denmark*

[3]*Orthopaedic Trauma Section, Department of Orthopaedic Surgery, Copenhagen University Hospital – Bispebjerg and Frederiksberg, Copenhagen, Denmark*

[4]*Brigham and Women's Hospital, Department of Orthopaedic Surgery, Harvard Medical School, Boston, USA*

Handling Editors: Paul Greenhaff & Koyal Garg

The peer review history is available in the Supporting Information section of this article (https://doi.org/10.1113/JP286507#support-information-section).

**Abstract figure legend** The interplay between current knowledge, treatment strategies and remaining unsolved clinical challenges in optimising clinical outcomes for patients suffering skeletal muscle tissue injury. In particular the outcome for strain injuries and acute extremity compartment syndrome remains poor, despite extensive work in animal models. A translational approach, bridging the gap between experimental models and clinical practice, involves examining treatment strategies in the surgical, rehabilitation and pharmacological realms. Created using Biorender.com.

**Abigail L. Mackey** is senior author and Clinical Professor in muscle physiology and regeneration after traumatic soft-tissue injury at the Institute of Sports Medicine Copenhagen. Her group investigates muscle regeneration, myofibre denervation and the myotendinous junction, in the context of ageing and exercise. **Grith Højfeldt**, postdoc, and **Christian Hoegsbjerg**, PhD student, are both part of Abigail s research group focusing on muscle injury and regeneration in the whole muscle and myotendinous junction respectively. **Arvind von Keudell** is an orthopaedic trauma surgeon at Bispebjerg Hospital, and Brigham and Women s Hospital. He serves as an associate professor at both Harvard Medical School and the University of Copenhagen.

The Journal of Physiology

**Abstract** Skeletal muscle injury and repair have been a major research focus for more than a century. Muscle injuries are defined by their cause and anatomical location and lie on a spectrum in terms of repair outcomes. From contraction-induced necrosis, which initiates regenerative myogenesis for complete restoration of tissue architecture and function to, at the other end of the spectrum, traumatic volumetric muscle loss (VML), where substantial portions (or the whole) of a muscle are lost, leaving the patient with permanent physical disability. Strain injuries are found between these two extremes and are characterised by healing with scar tissue formation and a high re-rupture rate. Across these injury types, a discriminating feature for a successful outcome is the preservation of the extracellular matrix (ECM) architecture of the muscle-tendon complex, in particular the myotendinous junction (MTJ). Numerous experimental models, imaging techniques and molecular analyses have led to a thorough understanding of how muscle stem cells interact with immune, vessel and stroma-associated cells during regenerative myogenesis. Paradoxically, treatment of muscle strain injury and VML has not improved, and regenerative engineering approaches remain a distant hope. Important issues for this field include matching the level of detail that exists for animal muscle regeneration with human data and identifying the site of tissue disruption during strain injury. We propose that a closer collaboration between cell biologists, physiologists, sports medicine practitioners and orthopaedic surgeons is required to improve patient outcomes, particularly for strain injuries and VML.

(Received 16 December 2024; accepted after revision 28 March 2025; first published online 28 April 2025)

**Corresponding authors** Grith Højfeldt and Abigail L. Mackey: Institute of Sports Medicine Copenhagen, Department of Orthopaedic Surgery, Copenhagen University Hospital - Bispebjerg and Frederiksberg, Bispebjerg Hospital (Building 8), Nielsine Nielsens Vej 11, Copenhagen NV, 2400 Denmark. Email: grith.stougaard.hoejfeldt@regionh.dk and abigailmac@sund.ku.dk

## Introduction

The largest cell of the body and main cell of skeletal muscle, the myofibre, is vital for overall health as it produces physical forces for breathing and movement. Additionally, myofibres play a crucial role in metabolic health through regulating glucose and lipid metabolism, and serving as the primary store of amino acids for protein synthesis for all tissues (Wolfe, 2006). Myofibre damage is, therefore, a threat to these essential functions and requires rapid and complete repair.

The longest myofibres in human muscles can contain tens of thousands of myonuclei per fibre (Hansson et al., 2020), which is an advantage for maintenance and growth throughout the fibre but also a challenge in situations of damage and repair. While the myofibre can self-repair minor damage induced by mild exercise (Roman et al., 2021), severely damaged myofibres undergo necrosis and need to be replaced. Satellite cells (Fig. 1), the resident stem cells on the surface of the muscle fibres, achieve this task through the process of regenerative myogenesis (Tajbakhsh, 2009). A feature of successful regeneration is preservation of the myofibre basement membrane and extracellular matrix (ECM) architecture of the muscle-tendon unit (Collins et al., 2024; Mackey & Kjaer, 2017a). However, not all muscle damage leads to myofibre necrosis, diminishing the importance of satellite cells in a range of muscle injury conditions. For example, in stark contrast to satellite cell-mediated regeneration lies volumetric muscle loss (VML), where a portion (or the whole) of a muscle, including the ECM, is permanently lost, leaving patients with significant physical disability. Similarly, athletes suffering strain injuries, where the muscle is separated from the tendon at the myotendinous junction (MTJ), remain plagued by a high re-rupture rate, indicating incomplete repair. In both situations, satellite cells are present but fail to restore tissue function due to the more serious degree of higher-level architectural disruption to the muscle-tendon complex.

To address these unsolved clinical challenges, muscle injury models have been developed, and intense effort has been invested in the field of tissue engineering and regenerative medicine in the last two decades. We now have unprecedented molecular-level insight into the processes of muscle regeneration, at least for the mouse – human data are lagging behind. Somewhat paradoxically, it can be argued that the clinical management of real-world muscle injuries and VML has not developed accordingly, raising critical questions about the relevance of current research models and highlighting the need for a paradigm shift in how we approach research on muscle repair. The purpose of this review is to combine insights from the basic science and clinical disciplines related to muscle repair in humans. We focus on different types

of injuries, each leading to markedly different outcomes in terms of regeneration and repair. These injuries range from experimental models to those encountered in real-world trauma and surgical settings. By examining these diverse situations collectively, we can identify areas of overlap and key knowledge gaps and questions that need to be addressed to improve clinical outcomes for patients affected by muscle loss and injury.

## The muscle injury spectrum

Muscle injuries are generally defined clinically according to the cause and anatomical site of injury. As such, muscle injuries are classified as either non-contact trauma, as seen with strains and intense repeated eccentric contractions, or direct contact, such as contusions or lacerations (Edouard et al., 2023; Garrett, 1990). However, the repair capacity following a muscle injury offers an equally relevant mode of distinction, that can be viewed on a spectrum (Fig. 2) and is largely determined by the severity of the injury, the site of tissue damage (and any associated complications such as age, disease state and comorbidities, which are beyond the scope of this review). The repair spectrum ranges from complete restoration of the tissue composition, as well as the architecture and function of the muscle-tendon complex, to, at the other end of the

spectrum, complete failure to restore any tissue structure and function. Between these two extremes lies healing by scar formation, which may seem unfavourable but should perhaps be viewed as an acceptable compromise in place of failed healing.

**Muscle injuries with complete regeneration.** The term muscle regeneration is often used in a broad sense. Here, we use the term regeneration to refer to adult regenerative myogenesis, defined as events that follow myofibre necrosis to result in myogenesis and new myofibre formation (Grounds, 2014), as opposed to minor damage self-repair by myonuclei (Roman et al., 2021). Myofibre necrosis triggers a complex biological response of three overlapping stages of regeneration (see Fig. 2; Fig. 3) (Hardy et al., 2016). Briefly, the initial phase is characterised by an inflammatory response driven by pro-inflammatory macrophages and then a shift towards anti-inflammatory macrophages (Saclier et al., 2013). In addition to their primary roles in clearing necrotic material, pro- and anti-inflammatory macrophages promote regeneration by stimulating quiescent satellite cells to progress through the myogenic programme of proliferation, differentiation and fusion (Saclier et al., 2013), which constitutes the middle phase of the regeneration process. During this process, the Pax7

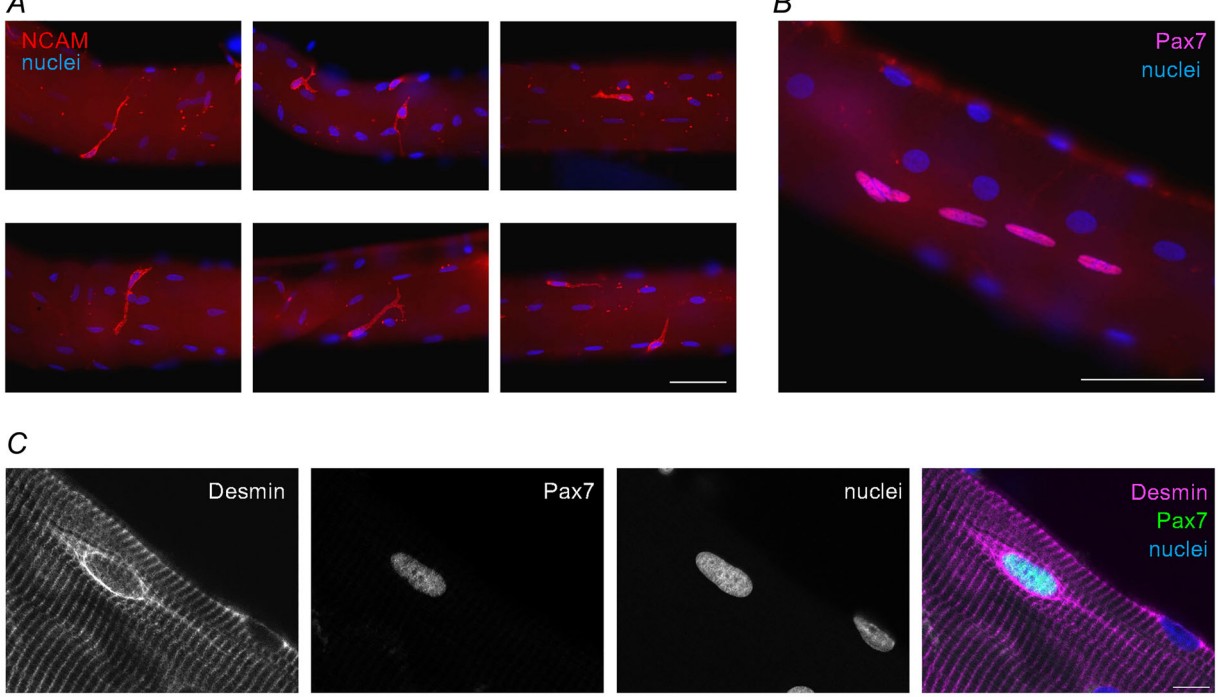

**Figure 1. Satellite cells on human muscle fibres**
*A* and *B*, widefield immunofluorescence of satellite cells on the surface of single human muscle fibres. Note the long cell lamellipodia when the satellite cells are stained with NCAM (*A*), compared to Pax7 (*B*), which is strictly confined to the nucleus. Scale bars represent 50 μm. *C*, confocal microscopy image of a satellite cell, expressing Pax7 and desmin, on the surface of a human muscle fibre collected 30 days after NMES. Scale bar, 10 μm.

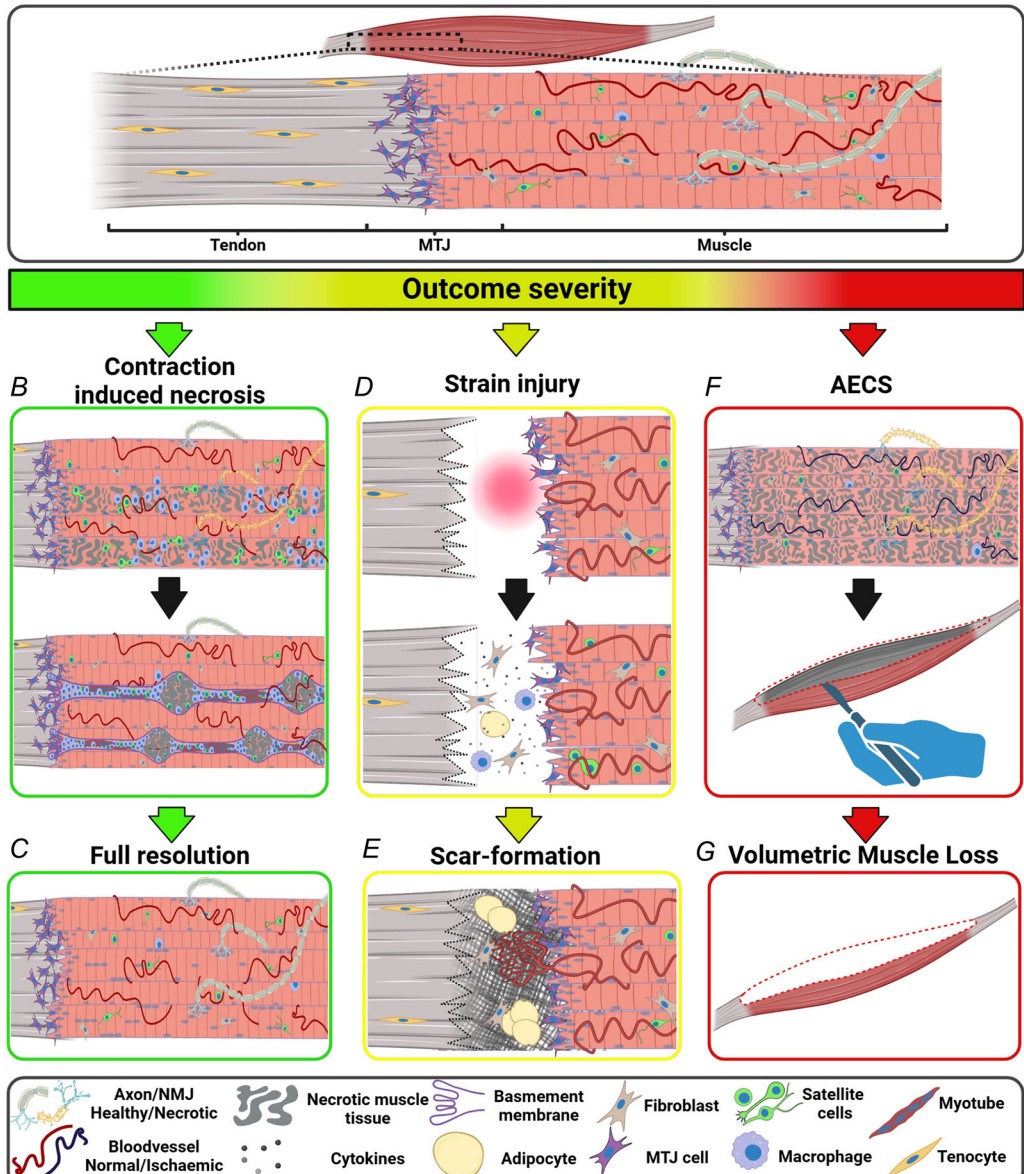

**Figure 2. The spectrum of human skeletal muscle regenerative capacity**
*A*, a schematic representation of the muscle at the cellular level, showing innervated myofibres firmly attached the tendon at the myotendinous junction (MTJ). *B* and *C*, contraction-induced necrosis. *B*, upon injury, such as extreme exercise involving repeated eccentric contractions, some myofibres will undergo full necrosis. At early stages of regeneration, macrophages infiltrate the tissue to phagocytose the necrotic tissue, while satellite cells proliferate and differentiate into myoblasts. Guided by the original and intact basement membrane, myoblasts fuse to form a new myofibre. *C*, in this injury model the myofibres are fully regenerated and the neuromuscular junction (NMJ) is re-established, with newly formed fibres typically displaying centralised myonuclei. It is currently unknown how this injury model affects the MTJ. *D* and *E*, strain injury. *D*, at the time of injury, the muscle separates from the tendon (or aponeurosis), allowing for the formation of a fluid-filled cavity, rich in inflammatory (immune cells and cytokines), adipogenic, and fibrotic markers. *E*, the strain injury ultimately fails to re-establish the original MTJ structure, and the muscle indirectly attaches to the tendon through a highly disorganised, vascularised and adipose scar tissue. *F* and *G*, acute extremity compartment syndrome (AECS). *F*, upon injury (e.g. bone fractures), the build-up of pressure within the muscle causes ischaemia, and myofibre damage and necrosis ensues. Due to the occluded capillaries, macrophages fail to infiltrate the tissue, so surgery is required to remove the excess necrotic tissue, leaving behind both undamaged and damaged, but viable, tissue. *G*, the surgical procedure leaves the patient with volumetric muscle loss, a permanent loss of muscle tissue. Not to scale. Created using Biorender.com.

*A*

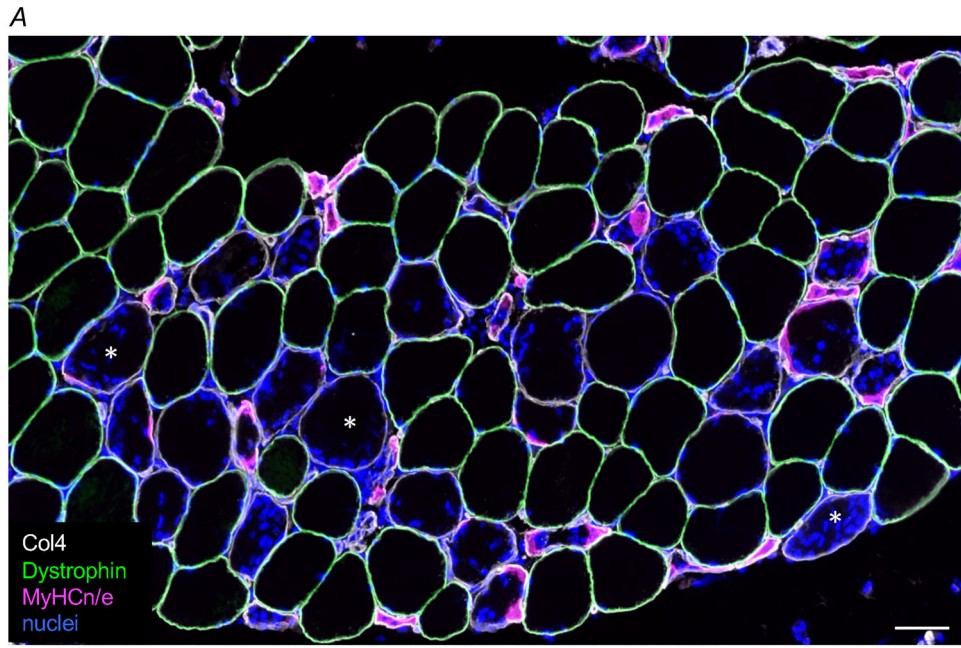

*B*

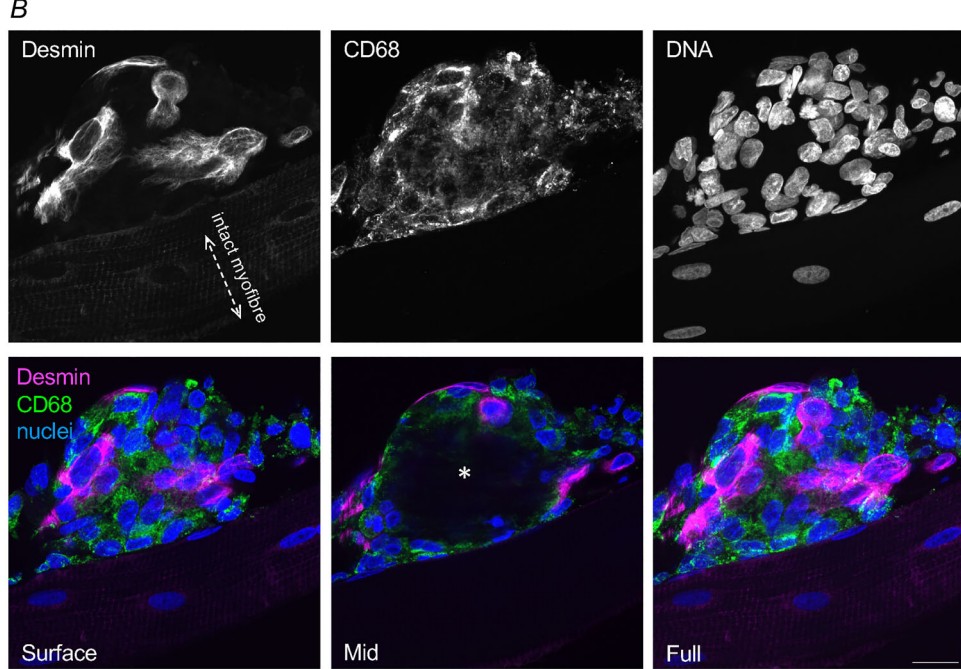

**Figure 3. Necrosis and regeneration of human skeletal muscle**

*A*, widefield immunofluorescence of a cross-section of human skeletal muscle collected 7 days after NMES, where signs of regeneration (neonatal/embryonic myosin; MyHCn/e) are visible inside the necrotic myofibres (lacking dystrophin; *example cells). Note the preservation of the basement membrane (collagen IV; col4) around the injured myofibres. Scale bar, 50 μm. *B*, confocal imaging of a single regenerating human muscle fibre (attached to an intact, uninjured fibre, below), where myogenic cells (desmin+) are seen together with macrophages (CD68+) on the surface of a necrotic area, corresponding to one of the areas marked with an asterisk (*) in *A*. An image slice through the mid-portion of the fibre reveals a necrotic core, void of cells. The greyscale single channel images, and full merged image, are maximum intensity projections of the full set of 18 images recorded in this stack. Scale bar, 20 μm.

expression and cell protrusions characteristic of the quiescent satellite cells (Ma et al., 2022), as seen in Fig. 1, are lost, in contrast to the constant expression of NCAM and appearance of other markers such as desmin (Figs 1 and 3). The third and final phase is the maturation of the newly formed myofibres (Ciciliot & Schiaffino, 2010).

Following injury, the integrity of the basement membrane is crucial for complete regeneration. As the myofibre undergoes necrosis, the basement membrane acts as a scaffold in the regenerative phase, ensuring the correct orientation of the newly formed myofibres (Collins et al., 2024). At this stage, centrally located myonuclei can be observed in both damaged (Murach et al., 2020) and new maturing fibres (Collins et al., 2024). Newly formed myofibres express developmental myosin (Fig. 3) (Ciciliot & Schiaffino, 2010), and rely on the re-establishment of connection with a motoneurone for full maturation. Without motoneurone input, the regenerating myofibres will remain small in size and will not be able to develop mature slow-type myosins and other sarcomeric proteins (Ciciliot & Schiaffino, 2010; Jerkovic et al., 1997). Once a neuromuscular connection is established, the fibre matures and can once again be recruited to contract and contribute to force production.

This process of regenerative myogenesis is triggered in humans by extreme exercise involving repeated eccentric contractions, often in a highly competitive setting (Marklund et al., 2013). Consequently, the most common model for studying muscle regeneration in humans is the use of repeated eccentric muscle contractions, usually with an isokinetic dynamometer. In the days after such a stimulus, indirect indications of muscle damage are evident, such as delayed onset muscle soreness (DOMS) and systemic increases in activity levels of the muscle enzyme creatine kinase (CK) (Mackey et al., 2004). However, analyses of muscle biopsies indicate that myofibre necrosis is rare in classical voluntary eccentric contraction studies with signs instead of small localised myofibre damage (Crameri et al., 2007; Roman et al., 2021; Yu et al., 2002). This is despite often profound changes in CK levels and, in biopsy specimens, clear evidence of stimulated activity of many cell types, as well as extensive remodelling of the muscle ECM (Crameri et al., 2004; Mackey & Kjaer, 2017b; Paulsen et al., 2013). Incidentally, it seems that voluntary eccentric contractions can induce necrosis in muscle groups for which this type of muscle contraction is unaccustomed, such as the elbow flexor biceps muscles (Paulsen et al., 2013), while the more commonly targeted quadriceps are exposed to eccentric contractions on a daily basis through walking and are therefore more difficult to damage. In a landmark study in 2007, Regina Crameri and colleagues at the Institute of Sports Medicine Copenhagen demonstrated that the use of neuromuscular electrical stimulation (NMES),

in conjunction with passive muscle lengthening by a dynamometer, represents a viable model for studying contraction-induced regenerative myogenesis. A bout of 200 of these NMES induced eccentric contractions is more effective at inducing myofibre necrosis than 200 voluntary eccentric contractions without NMES, in human quadriceps muscle (Crameri et al., 2007). This is probably due to the forced recruitment of motor units by NMES even after the myofibres and motoneurones become fatigued. Inducing myofibre necrosis, this model facilitates study of the entire *in vivo* myogenesis process in young and older individuals, from satellite cell activation to the maturation and fusion of new myotubes and myofibres (Crameri et al., 2007; Højfeldt et al., 2023; Karlsen et al., 2020; Mackey et al., 2016).

In humans, we are restricted by the number of muscle biopsies that can be taken, since a sample collected too close to a previous sampling site will be confounded by responses induced by the prior sampling (Vissing et al., 2005). In our experience, it is possible to collect four tissue samples with a 5–6 mm Bergstrom needle from the same vastus lateralis muscle without inducing satellite cell proliferation or necrosis in tissue collected at subsequent time points (days 2, 7 and 28 post NMES), by sampling perpendicular to the bone (and not angled in distal or proximal directions) and separating incision sites by at least 3 cm (Mackey et al., 2016). The same is seen in the biceps brachii, where repeated biopsies interspaced with 1–2 cm did not result in visible damage to the fibres sampled at later time points (Lauritzen et al., 2009). Taken together, the NMES model, when coupled with biopsy tissue sampling, represents a viable model for studying contraction-induced regenerative myogenesis in humans. With regard to the time course, rodent muscle regeneration has been described in more temporal detail than human. Species differences and commonalities are important to elucidate in order to evaluate the value of therapeutic targets identified in mouse models for human use.

While the muscle injuries just described can lead to complete recovery, providing a robust return to function, others result in the formation of scar tissue, which can complicate the healing process. Our current textbook understanding of muscle regeneration is largely based on rodent models of toxin injection or freeze injury (Hardy et al., 2016), which are difficult to translate to injury situations in humans. Importantly for myofibre attachment to the tendon, we lack insight into how the MTJ and the aponeurosis connections are reformed during regenerative myogenesis. With the replacement of the basement membrane (Mackey & Kjaer, 2017a), the myofibre tips need to re-attach to the tendon. Such insight may help understand the challenges of MTJ repair after strain injury, discussed next.

**Muscle injuries that lead to scar tissue formation.** Among the most common types of muscle injuries are strains, which occur when a muscle is overstretched or torn. Strains usually result from sudden movements involving high forces and are commonly observed in the hamstring muscles with activities like explosive sprinting (Edouard et al., 2023). Strain injuries can range from mild discomfort to severe tears, requiring extensive rehabilitation. Common for strains, and independent of severity, is that the majority of cases occurs at the MTJ. Depending on the context, the definition of the MTJ can vary. For example, it has been documented that for hamstring muscle strain injuries, 52% occur at the 'musculo-tendinous junction' and 18% at the 'musculo-aponeurotic junction' (Grange et al., 2023). For the purpose of this review, we refer to both of these anatomical regions under the term myotendinous junction, or MTJ.

Paradoxically, the MTJ is a highly adapted interface, with extensive membrane folding (Fig. 4), designed to withstand the demands of force transfer between two tissues of differing mechanical and structural properties and yet appears to be the weak link of the musculoskeletal chain (Mackey, 2025). Studies of strain injuries at the MTJ are sparse and limited to imaging modalities with insufficient power to resolve the site of tissue disruption at the ultrastructural level (Fig. 5). Most insights are, therefore, derived from animal models, which can have poor clinical translation, as discussed in detail elsewhere (Jakobsen & Krogsgaard, 2021).

In humans, strain injury studies are most often observational or epidemiological, focusing on injury type, location, recovery time and re-injury rates (Edouard et al., 2023; Grange et al., 2023). While providing important understanding in these domains, such studies cannot elucidate the underlying mechanisms of injury or repair. A few invasive human studies have, however, improved our understanding at the molecular level. For example, evidence points to a prolonged inflammatory response and an elevated presence of fibrotic markers and fibroblasts, indicative of excess matrix deposition, even up to 2 months after the initial trauma (Bayer et al., 2019). This is supported by the observations of scar tissue at the site of injury, observed by both MRI imaging (Silder et al., 2008) and microscopy of tissue samples collected months to years after injury (Bayer et al., 2021). Tissue alterations can also be seen along the length of the muscle, where thickening of the aponeurosis is observed (Bayer et al., 2021). It is interesting to note that while scar tissue (excess ECM at site of tissue injury) is a persistent and potentially permanent sign of prior injury, fibrosis (general thickening of existing ECM) does not develop with regeneration after contraction-induced necrosis (Mackey et al., 2016), again highlighting the fundamental difference in the pathology of, and repair

capacity of the muscle in response to, strain injury *versus* contraction-induced myofibre necrosis. The persistence of scar tissue after strain injury suggests that, in place of proper tissue healing, the healing process results in failed reconstruction of the MTJ, with muscle fibres attaching to an adipose and disorganised matrix of scar tissue (Bayer et al., 2021) instead of direct attachment to the tendon. Indeed, the formation of such an inferior muscle-scar-tendon complex could explain the high recurrence rates of muscle strain injuries (Wangensteen et al., 2016), raising the question of how targeted interventions can shift this response towards more effective regenerative outcomes.

Understanding the regenerative capacity and repair mechanisms of the MTJ at the cellular level has remained elusive. Developmental studies in rodents and birds have demonstrated that MTJ formation relies on the interplay between muscle- and tendon-derived cells (Esteves de Lima et al., 2021). Although the literature on regeneration of the mature MTJ is limited, animal studies have indicated that this interplay is preserved (Scott et al., 2019). The MTJ of both humans and mice appears to contain a large number of specialized myonuclei and tendon cells, transcriptionally distinct from their parent tissues, leading to the suggestion that these cells play an important role in the repair of the adult MTJ (Hoegsbjerg et al., 2025; Karlsen et al., 2023; Scott et al., 2019). However, the great capacity for repair observed in animal MTJ injury models (Scott et al., 2019) does not mimic muscle strain injury. This discrepancy may explain why the human MTJ appears to have a stunted regenerative capacity compared to the expectations set by animal models.

One of the major unresolved questions surrounding strain injury is the exact location and nature of tissue disruption within the MTJ. It has been suggested that muscle strain injuries, depending on the experimental setup, can occur at various points of the MTJ (as illustrated in Fig. 5): (1) the tendon attachment to the basement membrane, (2) the basement membrane attachment to the muscle fibre or (3) within the sarcomeres of the muscle fibre tip itself (Garrett et al., 1987; Tidball et al., 1993). Importantly, each of these sites is likely to require different repair mechanisms, highlighting the importance of determining the injury site for understanding both the injury progression and subsequent healing processes and how these can be supported through appropriate rehabilitation regimens, potentially along with pharmacological agents.

Another unresolved question is the necessity of the scar tissue formation. While scar formation is seen as incomplete recovery, with the muscle-tendon tissue not returning to the pre-injury state, this may very well be the favourable option. While strain injuries may lead to extensive rehabilitation and potential re-injury,

there is some degree of tissue re-attachment, albeit in the formation of scar tissue. In stark contrast, VML lies at the other end of the spectrum, characterized by the complete loss of muscle tissue and its ECM. This underscores a fundamental divergence in the healing processes of muscle injuries. While injuries following eccentric muscle contractions and strains may involve extensive inflammatory responses and localised repair mechanisms, VML showcases the complete failure of the remaining muscle to regenerate the lost muscle tissue, with potentially devastating outcomes for affected individuals.

**Muscle injury with no regeneration.** VML represents injuries where a part (or the whole) of a muscle is lost (Fig. 2). Since the ECM is also lost with VML, the lost muscle tissue cannot regenerate, and patients are left with significant physical disability. This type of muscle trauma

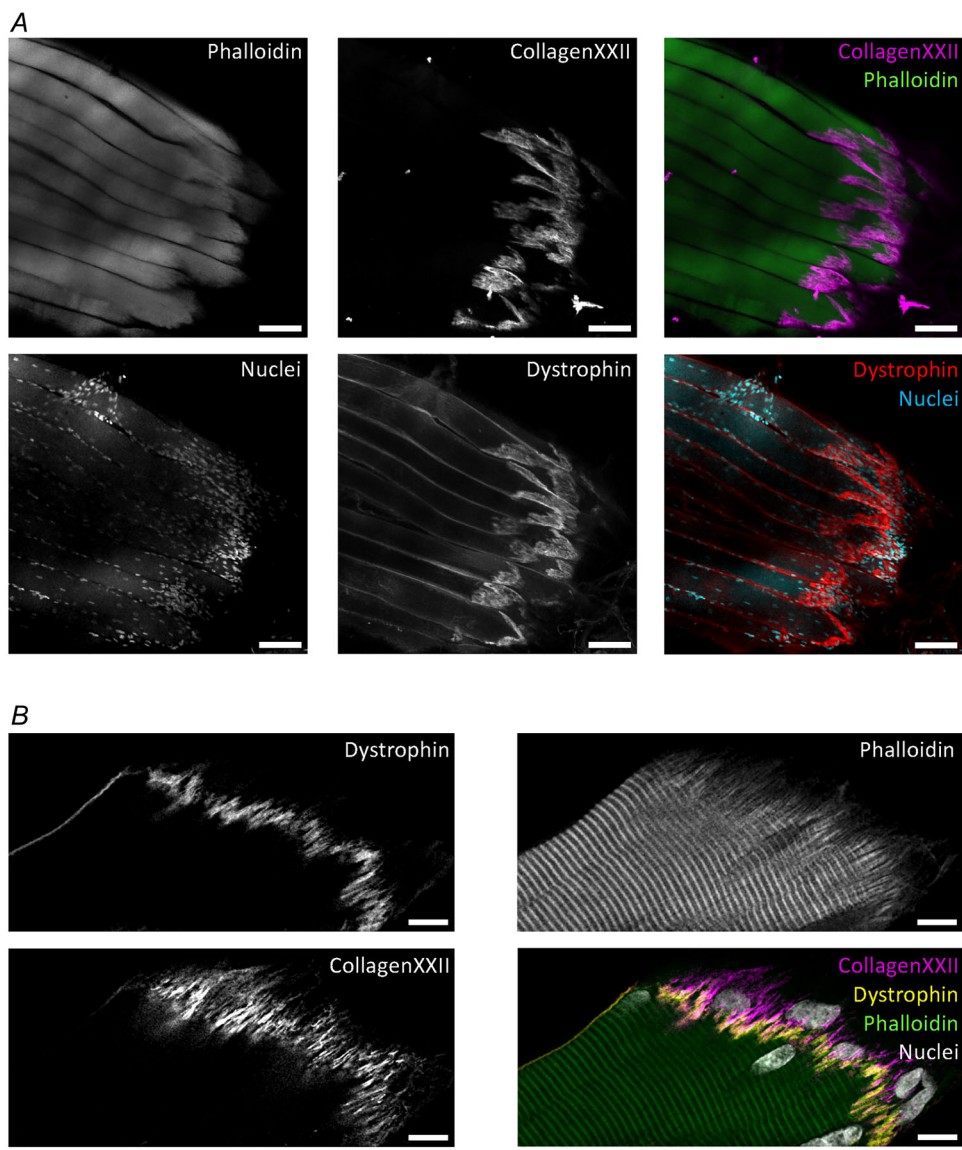

**Figure 4. Immunofluorescence of the human myotendinous junction**
*A*, the panels show single channel greyscale and composite confocal microscopy images of a bundle of human myofibres, with the myotendinous junction (MTJ) intact, stained for dystrophin (the sarcolemma), phalloidin (F-actin/sarcomeres), Hoechst (nuclei) and collagen XXII (MTJ basement membrane). Images are maximum intensity projections of *z*-plane image series spanning 22 μm. Note the aggregation of nuclei at the MTJ. Scale bar 100 μm. *B*, single slice confocal microscopy images of a single human myofibre stained together with the myofibre bundle in *A*. Here the folding of the MTJ is clearly visible, along with both myonuclei and nuclei of mononuclear cells closely associated with the MTJ. It is currently unknown at what ultrastructural level the human MTJ breaks in a strain injury, either the level of the myofibre (phalloidin), between dystrophin and the basement membrane (collagen XXII) or the tendon (see Fig. 5). Scale bars, 10 μm.

is sustained with traffic accidents or military combat, often due to bone fractures. Typically, the muscle is damaged directly by the same impact or object that fractured the bone, or indirectly where the sharp, jagged bone ends themselves inflict damage to the surrounding soft tissues. An associated condition, acute extremity compartment syndrome (AECS), exists where muscle damage begins insidiously, and if not treated can be fatal, although death is rare and disability is the most common outcome.

AECS following trauma is frequent, with an annual incidence of roughly 200,000 in the United states alone (Konstantakos et al., 2007). Of patients who sustained an extremity trauma, roughly 2.8% will require surgical intervention due to AECS, with the majority of incidents arising from open fractures, of which 5.9% require surgical intervention (Cone & Inaba, 2017). AECS develops due to pressure build up in a muscular compartment confined by rigid fascia, most commonly in the anterior compartment of the lower leg, home to the tibialis anterior and the deep peroneal nerve. Increasing pressure impairs tissue perfusion, and the resulting ischaemia leads to widespread myofibre damage and necrosis (Fig. 6). Further swelling ensues. This vicious cycle can only be stopped by releasing compartment pressure through emergency fasciotomy and removal of excess necrotic tissue, resulting in VML (Fig. 5), but also leaving behind some damaged muscle tissue that may regenerate. Without surgical intervention, there is increased risk of a greater extent of damaged muscle, greater disability, the need for amputation or in rare cases even death (von Keudell et al., 2015).

**Figure 5. Potential mechanisms of strain injuries**
*A*, the uninjured MTJ. *B–D*, potential injury mechanisms. *B*, a separation of the tendon, either at the basement membrane (BM) or in the adjacent tendon. This injury mode would leave the myofibres predominantly intact, potentially with the MTJ cells. *C*, a separation between the BM and the myofibres at the MTJ. This injury mode would see little direct myofibre damage but leave the myofibres separated from the MTJ cell population. *D*, disruption and separation within the myofibres. This injury mode would result in myofibre damage leading to focal injury repair, or necrosis of the entire myofiber, which would require regenerative myogenesis. Not to scale. Created using Biorender.com.

It is interesting to consider parallels between AECS and contraction-induced necrosis. In AECS, the muscle switches from victim to accomplice, since it is believed that the liquefied necrotic tissue enters the circulation and causes organ failure. The reason this does not happen with NMES and exercised-induced myonecrosis is that monocytes are free to exit the capillaries and infiltrate the muscle tissue to phagocytise the necrotic material. With AECS, however, the thinking is that the microvascular network is shut down due to intra-compartmental pressure, preventing monocytes from infiltrating the tissue. However, there must be other reasons for organ failure with AECS, since no organ complications have been seen following NMES despite circulating CK values of approximately 30,000–90,000 U/L peaking 4 days following NMES (Karlsen et al., 2020; Mackey et al., 2004, 2016), and myoglobin levels averaging 1048 μg/L on day 2 post NMES, with some individuals exceeding the assay upper detection limit of 3000 μg/L (Mackey et al., 2016). This high level of circulating breakdown products is seen without any concomitant complications, indicating good organ tolerance of muscle breakdown products in healthy young and older individuals, and at the same time raising questions about the aetiology of organ failure with AECS.

## Current treatment approaches and future perspectives

The capacity of skeletal muscle to regenerate is highly dependent on the type of injury and may be incomplete, and with or without scar tissue formation. Regenerative myogenesis appears to completely restore tissue architecture and function without specific rehabilitation

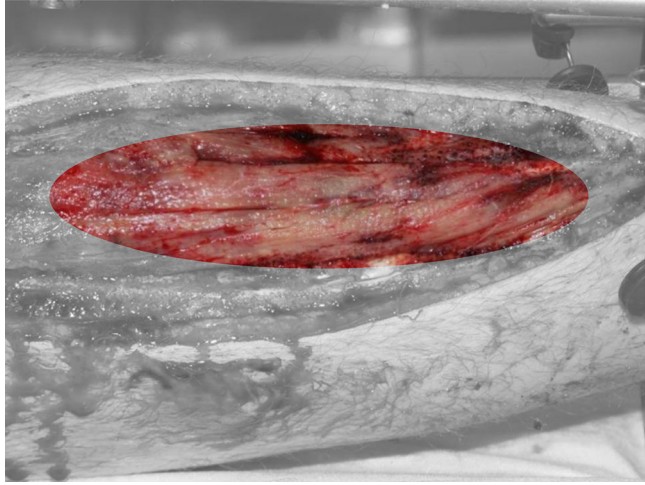

**Figure 6. Muscle necrosis with acute compartment syndrome**
The early onset of muscle necrosis with a dusky appearance (coloured inset) after surgical fasciotomy for a missed compartment syndrome in the tibialis anterior muscle.

programmes. On the other hand, the high re-injury risk after strain injury indicates incomplete repair. Current treatment strategies for muscle strain injuries continue to emphasise rehabilitative exercise, and only in rare cases require surgical treatment such as with complete quadriceps or hamstring tears. Another rare indication for the surgical treatment of strains is the evacuation of a large associated haematoma. These haematomas can cause severe pain and, in some rare instances, even compartment syndrome. With regard to rehabilitation, eccentric resistance exercises and rapid introduction to rehabilitation are both beneficial for return-to-sport outcomes (Bayer et al., 2017; Wulff et al., 2024). In combination with early mobilisation, there is now also an emphasis on the critical role of immune cell infiltration in the early phase of regeneration, leading to a new paradigm in acute injury treatment known as the 'PEACE & LOVE' approach, a holistic framework integrating protection, elevation, avoid anti-inflammatories, compression, educate, load, optimism, vascularisation, exercise (Dubois & Esculier, 2020). To develop more effective regenerative therapies, determining the precise site of failure at the MTJ is crucial. Once established, this knowledge would enable the development of more accurate injury models, that could guide the design of treatments targeting specific mechanisms of repair, whether through enhanced muscle fibre regeneration or facilitated re-attachment of muscle to the tendon or aponeurosis.

In both VML associated with fractures and muscle necrosis following AECS, clinical management focuses on mitigating immediate damage and addressing long-term functional deficits. For fractures with significant muscle loss, initial treatment involves surgical debridement and stabilisation of the fracture, often accompanied by the use of temporary wound management systems such as negative pressure therapy to preserve the tissue environment. However, definitive clinical treatment for VML remains limited and most often consists of muscle transfers to compensate for the loss of function, for example in the foot with severe VML after AECS (Chi et al., 2023). This treatment is not successful for all patients, however, highlighting the need for further understanding of how muscle transfers and regenerative medicine together can improve clinical outcomes for VML patients.

Tissue engineering and regenerative medicine have been around for several decades, and a common approach uses a decellularised ECM scaffold repopulated with tissue-specific relevant cells (Capella-Monsonis et al., 2024). Most effort is focused on generating synthetic or xenogeneic (from another species) scaffolds or bioengineered constructs for implanting surgically into areas of VML (Quarta et al., 2017). However, VML remains an unsolved clinical problem. Most experiments are still at the preclinical level in animals, with only a few

examples of implants into humans. Decellularised porcine bladder has been implanted into the tibialis anterior of humans with some, though limited, success (Sicari et al., 2014). The main challenge is rejection of the implant by the host immune system and a lack of improved function due to failed innervation of the newly formed muscle fibres. Autologous (from self) cells and scaffolds, therefore, represent an alternative solution, although the process of decellularising the ECM and repopulating it with cells is arduous and time-consuming (Lu et al., 2013) and would involve several steps for the patient and surgeon. An alternative is the rather old concept of minced muscle fragments which leads to some degree of muscle regeneration (Carlson & Gutmann, 1972; Studitsky, 1964) and due to its relatively simple procedure is again gaining interest (Aguilar et al., 2018).

Tissue engineering approaches are also beginning to focus on the MTJ (Shama et al., 2024). Much of this work has focused on developing entirely new MTJ reconstructions. However, it remains challenging to see how such constructs could be integrated into strain-injured tissue. Nonetheless, engineered MTJ constructs serve as valuable *in vitro* models for testing regenerative therapies. For example, specialised cellular populations at the MTJ (Scott et al., 2019) have been suggested to improve MTJ repair outcomes and thus represent a potential target for cellular therapies in muscle strain injuries. A potential therapeutic strategy could involve combining cellular therapies with scaffolds designed to guide and organise endogenous repair processes, minimising scarring and enhancing restoration of physical function. However, these options remain theoretical until the site of tissue failure is determined at the ultrastructural level.

Future research should adopt a multidisciplinary approach, but most importantly, would benefit from closer ties between basic science and clinical practice to ensure precise injury models that more closely represent the clinical cases. This will be a critical step to ensure that scientific advances translate into therapeutic advancements that benefit human patients.

## Conclusions

Injuries to human skeletal muscle span a wide spectrum, from complete regeneration to permanent functional deficits. While great advancements have been made in molecular-level understanding of rodent muscle regeneration, equivalent human data are lacking and species differences in the time course and processes of regeneration remain to be clarified if therapeutic targets identified in mouse models are to be considered for use in humans. Furthermore, an improved understanding of the molecular and structural composition of the MTJ and

its breaking point are vital for progression in the clinical management of strain injuries. A greater awareness of the relevance of experimental injury models will provide more specific and effective post-injury recovery regimens for the respective type of muscle injury, ultimately leading to improved clinical outcomes for patients across the full spectrum of muscle injury.

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

## Additional information

### Data availability statement

Not applicable.

### Competing interests

None.

### Author contributions

All authors contributed to the conception and drafting of the work and revising it critically for important intellectual content. All authors have approved the final version of the manuscript and agree to be accountable for all aspects of the work in ensuring that questions related to the accuracy or integrity of any part of the work are appropriately investigated and resolved. All persons designated as authors qualify for authorship, and all those who qualify for authorship are listed.

### Funding

Lundbeck Foundation (Grant ID: R344-2020-254 to A.L.M.), Independent Research Fund Denmark (Grant ID:

10.46540/3101-00063B to A.L.M.), and the BRIDGE - Translational Excellence Programme (bridge.ku.dk) at the Faculty of Health and Medical Sciences, University of Copenhagen, funded by the Novo Nordisk Foundation (Grant ID: NNF20SA0064340 to G.H.).

## Supporting information

Additional supporting information can be found online in the Supporting Information section at the end of the HTML view of the article. Supporting information files available:

**Peer Review History**

## Keywords

acute extremity compartment syndrome, myonecrosis, myotendinous junction, regenerative myogenesis, scar tissue, strain injury, tissue engineering, volumetric muscle loss

