## [Peer Review History · The Journal of Physiology]

The repair capacity spectrum of human skeletal muscle injury from sports to surgical trauma settings

Grith Højfeldt, Christian Hoegsbjerg, Arvind G. von Keudell, and Abigail Louise Mackey
DOI: 10.1113/JP286507

Corresponding author(s): Abigail Mackey (abigailmac@sund.ku.dk)

The following individual(s) involved in review of this submission have agreed to reveal their identity: Ferdinand von Walden (Referee #1); Baptiste Jude (Referee #2)

Review Timeline:

Submission Date:	16-Dec-2024
Editorial Decision:	27-Jan-2025
Revision Received:	01-Mar-2025
Accepted:	28-Mar-2025

Senior Editor: Paul Greenhaff

Reviewing Editor: Koyal Garg

Transaction Report:

Dear Professor Mackey,

Re: JP-TR-2024-286507 "The repair capacity spectrum of skeletal muscle after injury in sports and surgical trauma settings" by Grith Højfeldt, Christian Hoegsbjerg, Arvind G. von Keudell, and Abigail Louise Mackey

Thank you for submitting your manuscript to The Journal of Physiology. It has been assessed by a Reviewing Editor and by 2 expert referees and we are pleased to tell you that it is acceptable for publication following satisfactory revision.

ABSTRACT FIGURES: Authors may use The Journal's premium BioRender account to create/redraw their Abstract Figures (and any other suitable schematic figure). Information on how to access this account is here: <https://physoc.onlinelibrary.wiley.com/journal/14697793/biorender-access>.

REVISION CHECKLIST: Upload a full Response to Referees file. To create your 'Response to Referees' copy all the reports, including any comments from the Senior and Reviewing Editors, into a Microsoft Word, or similar, file and respond to each point, using font or background colour to distinguish comments and responses and upload as the required file type.

We look forward to receiving your revised submission.

Yours sincerely,

Paul Greenhaff
Senior Editor

EDITOR COMMENTS

Reviewing Editor:

The manuscript can proceed to publication following adequate revisions, including improving the connections between sections, which presently feel disjointed.

Senior Editor:

Thank you for submission of this Topical review article that has been considered by a reviewing editor and two expert reviewers. All are of the opinion that the manuscript is well written and will be of interest to the readership of The Journal of Physiology. Nevertheless, both reviewers believe that some aspects of the manuscript could be improved. For example, more nuanced discussion of volumetric muscle loss, especially in the context of Acute Extremity Compartment

Syndrome that most likely includes both muscle fibre necrosis and regeneration. Secondly, better justification of the sub-classes of injuries detailed by the authors. We look forward to receiving the revised version of the review.

REFEREE COMMENTS

Referee #1:

The review by Hojfeldt and co-workers is focused on a hot and highly relevant topic to the readership of Journal of Physiology. The manuscript is well written and easy to read. The figures in general and especially the immunohistochemistry is top level and could be used more in the text in my opinion.

Detailed comments per section

Abstract - I think the the word "removed" in conjunction with VML implies that skeletal muscle can only lost as a result of surgical debridement. What about traumatic VML? Could this be rewritten to include a wider ethiology of VML?

Intro

What is the evidence for whole fiber necrosis as a result of a full but local fiber damage? For example, following skeletal muscle biopsies in the VL, repeated biopsies days/weeks after does not typically include signs of necrotic/regenerating fibers 2-3 cm away from the previous biopsy? Is this due to the pennate nature of the VL? Would the result be different in the biceps for example?

Muscle injury spectrum

Please consider discussion the role of age and co-morbidities whit respect to capacity for repair?

Muscle injuries with complete regeneration

When referring to Fig 2 and AECS, I would suggest recognizing that it most likely involves other types of regeneration too? The muscle that is not surgically removed is most likely subject to regeneration.

When refering to new maturing fibers, I believe that at least the mouse literature recognizes that resident myonuclei can

become centrally located during regeneration. It's not 100% newly formed fibers. Please review work by Kevin Murach.

Is motoneurons correctly spelled? Please make sure that "in vivo" and similar expressions are in italic.

Could you please be a bit more detailed with respect to the number of voluntary contractions needed to induce necrosis in unaccustomed skeletal muscles?

I would suggest to be specific that NEMS coupled with voluntary Excentric actions represents a viable model for studying contraction-induced regenerative myogenesis.

Muscle injuries that lead to scar tissue formation

Please consider discussing how the different types of strain-injuries can be treated by surgical repair. Which type do you think is best suited for surgery?

Muscle injury with no regeneration

I think it's more common that AECS results in impaired function as compared to being fatal. Please consider tempering the first statement.

Please be more detailed with respect to what type of injury predisposes an individual for AECS. As it currently reads, one could get the impression that 10% of all lower leg fractures will result in AECS which I think is a gross overstatement.

Any information on myoglobin levels or potassium levels in the Mackey papers that you are referring to with NEMS?

Referee #2:

This is a very interesting review pointing out some specificities of the human skeletal muscle regeneration/repair with a focus on damages induced by exercise and trauma.

The authors presented nicely in the introduction the muscle repair in different conditions. However, some connections are missing. Indeed, the title presents muscle repair in sports injury (with the complete regeneration of the fiber or scar formation at the NMJ) and surgical trauma, and the connection between both is missing. Moreover, in the first part on the complete regeneration, it might be interesting to also talk about the local micro damages in the fibers, as explained in the introduction, that will not require the sat cell activation but still is an important component of the muscle regeneration. Furthermore, in the second paragraph, the focus is on strains and their impact on MTJ. Again, this is very interesting and less common, but I would appreciate an opening on the topic of scar formation or a justification why we are only talking about the NMJ. Scar tissue can't occur in the middle of the tissue for example? This is a real question that I would like to see at the beginning of the paragraph, to justify why the authors focus only on MTJ. For the last part and the VML, I understand the model with no regeneration but I miss a transition since all the review so far focus on exercise-inducing muscle damage. There are other models inducing muscle damage/repair, as discussed in the review with some limitations, but I miss a transition. If you want to talk about a models inducing muscle damage in that case I would like for the other parts some examples of scar tissue formation, related to inflammation and/or pathologies such as DMD or CP are known to have fibrotic muscle for example. All these questions are well investigated by the lab for years now but just a bit more of a connection would make this review excellent.

For a few more points:

I would add "Human" in the title since the review focuses almost only on human muscle regeneration.

Page 1: Arvind G von Keudell affiliations seem to be 3 and 4 instead.

I would use fiber instead of fibre.

Page 4: "Following maturation, the myofibre relies on re-establishment of connection with a motoneurone for full maturation into fiber type-specific myosins and other sarcomeric proteins". I would highlight a little more the impact of MN connection with MyHC switch since it's a key step in the transition from the regenerative to the mature fiber, and the nerve stimulation will mainly drive the phenotype of the fiber.

Page 5: I would like to see the difference between fibrotic and scar tissue if there is any.

The figures are really nice but not used enough. Fig1, the difference between NCAM and Pax7 staining is beautiful but not used by the authors. Also, for panel C, I would prefer to see the colors instead of black and white. Same for all the other figures. Fi2, nice overview of the review but you wrote page 4 "muscle biopsies indicate that myofiber necrosis is rare". What about micro damages?

REQUIRED ITEMS

- Please include an Abstract Figure file, as well as the Figure Legend text within the main article file. The Abstract Figure is a piece of artwork designed to give readers an immediate understanding of the Review Article and should summarise the main conclusions. If possible, the image should be easily 'readable' from left to right or top to bottom. It should show the physiological relevance of the Review so readers can assess the importance and content of the article. Abstract Figures should not merely recapitulate other figures in the Review. Please try to keep the diagram as simple as possible and without superfluous information that may distract from the main conclusion of the Review. Abstract Figures must be provided by authors no later than the revised manuscript stage and should be uploaded as a separate file during online submission labelled as File Type 'Abstract Figure'. Please ensure that you include the figure legend in the main article file. All Abstract Figures will be sent to a professional illustrator for redrawing and you may be asked to approve the redrawn figure before your paper is accepted.

- Please upload separate high quality figure files via the submission form.

- Author profile(s) must be uploaded via the submission form. Authors should submit a short biography (no more than 100 words for one author or 150 words in total for two authors) and a portrait photograph of the two leading authors on the paper. These should be uploaded and clearly labelled together in a Word document with the revised version of the manuscript. Any standard image format for the photograph is acceptable, but the resolution should be at least 300 DPI and preferably more. A group photograph of all authors is also acceptable, providing the biography for the whole group does not exceed 150 words.

- Please include a full title page as part of your main article (Word) file, which should contain the following: title, authors, affiliations, corresponding author name and contact details, keywords, and running title.

- Please ensure that the Article File you upload is a Word file.

END OF COMMENTS

EDITOR COMMENTS

Reviewing Editor:

The manuscript can proceed to publication following adequate revisions, including improving the connections between sections, which presently feel disjointed.

Senior Editor:

Thank you for submission of this Topical review article that has been considered by a reviewing editor and two expert reviewers. All are of the opinion that the manuscript is well written and will be of interest to the readership of The Journal of Physiology. Nevertheless, both reviewers believe that some aspects of the manuscript could be improved. For example, more nuanced discussion of volumetric muscle loss, especially in the context of Acute Extremity Compartment Syndrome that most likely includes both muscle fibre necrosis and regeneration. Secondly, better justification of the sub-classes of injuries detailed by the authors. We look forward to receiving the revised version of the review.

We thank the Reviewing Editor and Senior Editor for handling our manuscript and for the opportunity to revise it. We have addressed all the reviewer comments in a point-by-point fashion, paying extra attention to justifying the sub-classes of injuries we cover as well as the transitions between them throughout the manuscript. We have done this by generating an abstract figure, and adding extra text at the points where we transition to another injury type. Furthermore, we appreciate the points about muscle regeneration in disease states but prefer to keep the original focus on the healthy individual and the physiological mechanisms involved in muscle repair rather than introducing confounding factors involved in genetic diseases, which would not only bring us over the word and reference limit but which we feel would be better suited to a different review and perhaps another journal.

REFEREE COMMENTS

Referee #1:

We would like to thank both reviewers for their work reviewing our manuscript. We have provided point-by-point responses below (referencing line numbers in the clean version of the revised manuscript).

The review by Hojfeldt and co-workers is focused on a hot and highly relevant topic to the readership of Journal of Physiology. The manuscript is well written and easy to read. The figures in general and especially the immunohistochemistry is top level and could be used more in the text in my opinion.

We would like to thank Referee #1 for the time and effort spent reviewing our manuscript. We have referred to our immunohistochemistry images more frequently throughout the manuscript to make better use of them.

Detailed comments per section

Abstract - I think the word "removed" in conjunction with VML implies that skeletal muscle can only be lost as a result of surgical debridement. What about traumatic VML? Could this be rewritten to include a wider etiology of VML?

This is a valid point and we have changed the sentence to reflect both that muscle tissue can be lost due to surgical debridement and the trauma incident itself. We have changed the word "removed" to "lost" in the abstract and added the word "traumatic", so it now reads:

"From contraction-induced necrosis, which initiates regenerative myogenesis for complete restoration of tissue architecture and function to, at the other end of the spectrum, traumatic volumetric muscle loss (VML), where substantial portions (or the whole) of a muscle are lost, leaving the patient with permanent physical disability."

Intro

What is the evidence for whole fiber necrosis as a result of a full but local fiber damage? For example, following skeletal muscle biopsies in the VL, repeated biopsies days/weeks after does not typically include signs of necrotic/regenerating fibers 2-3 cm away from the previous biopsy? Is this due to the pennate nature of the VL? Would the result be different in the biceps for example?

As we understand the question, the reviewer would like to know if a local fibre damage insult, such as taking a needle biopsy, can cause necrosis of the entire muscle fibre. This is a good question which is not that straightforward to answer. However, there are some clues. For example, in the control leg data we do not see any necrosis or increase in satellite cell number in the VL following biopsies collected on days 2, 7 and 28 post NMES when interspaced by 2-3 cm along the VL (Mackey *et al.*, 2016). This indicates that our repeated biopsies are not collected by previous sampling, when sampled the way we do it. Likewise, Lauritzen *et al.* do not see any necrosis in the biceps following repeated biceps biopsies interspaced by 1-2 cm (Lauritzen *et al.*, 2009), again supporting our data. But this does not mean that biopsy sampling does not cause necrosis of the entire muscle fibre – it may well do which is why we try to avoid previous sampling sites. The reason we succeed is likely explained by the reviewer's point about the pennate nature of both the VL and the biceps muscles.

We realise this comment was in relation to the introduction section but have added the new text to a later section (paragraph starting at Line 160), where we thought it fit in well with existing detail on repeated biopsies:

"In our experience, it is possible to collect four tissue samples with a 5-6mm Bergstrom needle from the same vastus lateralis muscle without inducing satellite cell proliferation or necrosis in tissue collected at subsequent time points (days 2, 7, and 28 post NMES), by sampling perpendicular to the bone (and not angled in distal or proximal directions) and separating incision sites by at least 3cm (Mackey *et al.*, 2016). The same is seen in the biceps brachii, where repeated biopsies interspaced with 1-2 cm did not result in visible damage to the fibres sampled at later time points (Lauritzen *et al.*, 2009)."

Muscle injury spectrum

Please consider discussion the role of age and co-morbidities whit respect to capacity for repair?

It is well taken that there are many factors that can influence repair capacity. Age and comorbidities are certainly important and have now been mentioned, although only briefly, as there are also data for example showing that older muscle can regenerate just as well as young (Karlsen *et al.*, 2020). As much as possible, we wanted to restrict the scope of this review to healthy individuals. Also, introducing genetic disease and comorbidities would require extensive addition of text and we are already at the limit regarding space and number of references. However to recognise this point, we have added the following text, starting on Line 90:

“However, the repair capacity following a muscle injury offers an equally relevant mode of distinction, that can be viewed on a spectrum (Figure 2) and is largely determined by the severity of the injury, the site of tissue damage (and any associated complications such as age, disease state, and comorbidities, which are beyond the scope of this review).”

Muscle injuries with complete regeneration

When referring to Fig 2 and AECS, I would suggest recognizing that it most likely involves other types of regeneration too? The muscle that is not surgically removed is most likely subject to regeneration.

This is a very good point and we agree that the widespread ischemia within the muscle is also very likely to cause damage to the muscle which is not removed. We have therefore rewritten the figure legend for Figure 2 to highlight this. It now reads:

“F-G: Acute extremity compartment syndrome (AECS). F) Upon injury (e.g. bone fractures), the build-up of pressure within the muscle causes ischaemia, and myofibre damage and necrosis ensue. Due to the occluded capillaries, macrophages fail to infiltrate the tissue, so surgery is required to remove the excess necrotic tissue, leaving behind both undamaged and damaged, but viable, tissue. G) The surgical procedure leaves the patient with volumetric muscle loss, a permanent loss of muscle tissue.”

We have also rewritten a segment from the section “muscle injury with no regeneration” so it reads (starting at Line 284):

“Increasing pressure impairs tissue perfusion, and the resulting ischaemia leads to widespread myofibre damage and necrosis (Figure 6). Further swelling ensues. This vicious cycle can only be stopped by releasing compartment pressure through emergency fasciotomy and removal of excess necrotic tissue, resulting in VML, but also leaving behind some damaged muscle tissue that may regenerate. Without surgical intervention, there is increased risk of greater extent of damaged muscle, greater disability, the need for amputation, or in rare cases even death (von Keudell *et al.*, 2015).”

When refering to new maturing fibers, I believe that at least the mouse literature recognizes that resident myonuclei can become centrally located during regeneration. It's not 100% newly formed fibers. Please review work by Kevin Murach.

Thank you for bringing this intriguing finding to our attention. We have now referenced the work by Kevin Murach and colleagues (Murach *et al.*, 2020), which we believe the reviewer is referring to. We have also added this to the text at Line 122 so it includes this distinction:

“At this stage, centrally located myonuclei can be observed in both damaged (Murach *et al.*, 2020) and new maturing fibres (Collins *et al.*, 2024).”

Is motoneurons correctly spelled? Please make sure that "in vivo" and similar expressions are in italic.

Thank you for your attention to detail. The spelling throughout is based on British spelling, according to the preference of the journal. In British English, motoneurones is the correct spelling.

“in vivo” and “in vitro” are now in italics throughout.

Could you please be a bit more detailed with respect to the number of voluntary contractions needed to induce necrosis in unaccustomed skeletal muscles?

We have added that the protocol we have used involves approximately 200 contractions, which we know works. We do not know what the minimum required number of contractions is, as we have not done a dose response relationship. In principle this could be done, but given the large interindividual variability we observe in the extent of damage in response to 200 contractions, a large number (100+) of participants would be required to determine the minimum number required to induce necrosis – it could even be individual where 10 contractions might suffice for some (if untrained enough and an effective electrical stimulation can be achieved), but this is very speculative.

We have clarified the number of contractions starting at Line 150:

“A bout of 200 of these NMES induced eccentric contractions is more effective at inducing myofibre necrosis than 200 voluntary eccentric contractions without NMES, in human quadriceps muscle”

I would suggest to be specific that NEMS coupled with voluntary Excentric actions represents a viable model for studying contraction-induced regenerative myogenesis.

We have clarified that the movements we refer to in the Crameri *et al.* 2007 study are not voluntary. With this model, electrical stimulation forces the muscle to contract, while a dynamometer passively bends the knee, causing a lengthening, and thus eccentric contraction, of the vastus lateralis. The text now reads (Line 148):

“the use of neuromuscular electrical stimulation (NMES), in conjunction with passive muscle lengthening by a dynamometer, represents a viable model for studying contraction-induced regenerative myogenesis.”

There is, to our knowledge, no data showing that this can also be achieved with voluntary contractions coupled with NMES (although we don't see why it should not be just as effective, but perhaps more challenging to co-ordinate the timing of the voluntary and stimulated contractions).

Muscle injuries that lead to scar tissue formation

Please consider discussing how the different types of strain-injuries can be treated by surgical repair. Which type do you think is best suited for surgery?

Strain injuries are rarely treated surgically, but we agree that it is helpful to mention when they are. We have added the following to the “Current treatment approaches and future perspectives” section (Line 314):

“...and only in rare cases require surgical treatment such as complete quadriceps or hamstring tears. Another rare indication for the surgical treatment of strains is the evacuation of a large associated haematoma formation. These haematomas can cause severe pain and, in some rare instances, even compartment syndrome. “

Muscle injury with no regeneration

I think it's more common that AECS results in impaired function as compared to being fatal. Please consider tempering the first statement.

It is true that death is not the likely outcome of AECS, particularly if treated. However, if untreated there is a non-negligible risk of death, with the mortality rate ranging from 5-47% depending on the study (although the high mortality rate studies have multiple traumas and co-morbidities compounding the effect of AECS). We therefore still see it fitting to mention death as a possible outcome.

We have tempered the following two sentences from this section on Lines 275 and 290:

“An associated condition exists where muscle damage begins insidiously, and if left untreated may be fatal – Acute Extremity Compartment Syndrome (AECS), although death is rare and disability is the most common outcome.”

Without surgical intervention, there is increased risk of greater extent of damaged muscle, greater disability, the need for amputation, or in rare cases even death (von Keudell et al., 2015).

Please be more detailed with respect to what type of injury predisposes an individual for AECS. As it currently reads, one could get the impression that 10% of all lower leg fractures will result in AECS which I think is a gross overstatement.

This is a fair point. We have now been more specific about the numbers we present from Konstantakos et al, and Cone & Inaba, and have re-written the section so it reads (starting on Line 278):

“AECS following trauma is frequent, with an annual incidence of roughly 200,000 in the United States alone (Konstantakos et al., 2007). Of patients who sustained an extremity trauma, roughly 2.8% will require surgical intervention due to AECS, with the majority of incidents arising from open fractures, of which 5.9% require surgical intervention (Cone & Inaba, 2017).”

Any information on myoglobin levels or potassium levels in the Mackey papers that you are referring to with NEMS?

Yes, the Mackey FASEB paper from 2016 did measure myoglobin. We have added the values to the manuscript on Line 302:

...“and myoglobin levels averaging 1048 $\mu\text{g/l}$ on day 2 post NMES, with some individuals exceeding the assay upper detection limit of 3000 $\mu\text{g/l}$ (Mackey et al., 2016).”

Referee #2:

We would like to thank both reviewers for their work reviewing our manuscript. We have provided point-by-point responses below (referencing line numbers in the clean version of the revised manuscript).

This is a very interesting review pointing out some specificities of the human skeletal muscle regeneration/repair with a focus on damages induced by exercise and trauma.

The authors presented nicely in the introduction the muscle repair in different conditions. However, some connections are missing. Indeed, the title presents muscle repair in sports injury (with the complete regeneration of the fiber or scar formation at the NMJ) and surgical trauma, and the connection between both is missing. Moreover, in the first part on the complete regeneration, it might be interesting to also talk about the local micro damages in the fibers, as explained in the introduction, that will not require the sat cell activation but still is an important component of the muscle regeneration.

Thank you for your constructive comments and suggestions. Your point regarding the connection between the injuries (and thus the review) is well taken. We have made some alterations to the introduction, and the introduction to the section “The muscle injury spectrum” to address the connection between the different injury types. Furthermore, the new abstract figure aims to combine the different injury types under the common umbrella of optimising translational approaches to improve patient outcomes in relation to the many types of injury that can be sustained by skeletal muscle. Of course there are other types of injury. Our goal though was to select a few at the extreme ends of the repair capacity spectrum as well as in between (also including micro damage), to provide examples for discussing specific mechanisms and clinical outcomes. We hope the reviewer finds the revised version satisfies this point.

Furthermore, in the second paragraph, the focus is on strains and their impact on MTJ.

Again, this is very interesting and less common, but I would appreciate an opening on the topic of scar formation or a justification why we are only talking about the NMJ. Scar tissue can't occur in the middle of the tissue for example? This is a real question that I would like to see at the beginning of the paragraph, to justify why the authors focus only on MTJ.

This is a good point, that we should have made clearer. In this review we refer to injuries where myofibres attach to tendinous tissue, which includes both the “musculo-tendinous junction” (where 52% of hamstring injuries occur, Grange et al. 2023) and the “musculo-aponeurotic junction” (where 18% of hamstring injuries occur, Grange et al. 2023), together make up 70% of all hamstring muscle strain injuries (based on 2761 injuries in 34 studies; Grange et al. 2023). We have now clarified in the paragraph starting at Line 187 that we consider the aponeurosis included in the term myotendinous junction (MTJ, which is what we understand the reviewer is referring to, and not the NMJ):

“Depending on the source, the definition of the MTJ can vary. For example it has been documented that for hamstring muscle strain injuries, 52% occur at the “musculo-tendinous junction” and 18% at the “musculo-aponeurotic junction” (Grange et al., 2023). For the purpose of this review, we refer to both of these anatomical regions under the term myotendinous junction, or MTJ.”

For the last part and the VML, I understand the model with no regeneration but I miss a transition since all the review so far focus on exercise-inducing muscle damage. There are other models inducing muscle damage/repair, as discussed in the review with some limitations, but I miss a transition. If you want to talk about a models inducing muscle damage in that case I would like for the other parts some examples of scar tissue formation, related to inflammation and/or pathologies such as DMD or CP are known to have fibrotic muscle for example. All these questions are well investigated by the lab for years now but just a bit more of a connection would make this review excellent.

We appreciate the value of including some diseases such as DMD or CP in our review but since we are already over the limit for text and number of references, we cannot expand the scope. The initial idea was to restrict the focus to healthy individuals as much as possible since the molecular processes driving for example fibrosis may vary between the healthy and diseased state and with many confounding factors the discussion would become quite speculative and be better suited for another journal. To recognise this point though, we have added the following text, starting on Line 90:

“However, the repair capacity following a muscle injury offers an equally relevant mode of distinction, that can be viewed on a spectrum (Figure 2) and is largely determined by the severity of the injury, the site of tissue damage (and any associated complications such as age, disease state, and comorbidities, which are beyond the scope of this review).”

For a few more points:

I would add "Human" in the title since the review focuses almost only on human muscle regeneration.

We have added “human” to the title.

Page 1: Arvind G von Keudell affiliations seem to be 3 and 4 instead.

Thank you for pointing this out. You are correct, and this is changed.

I would use fiber instead of fibre.

The spelling throughout the manuscript is based on British spelling, as per the journal guidelines, so we prefer to keep “fibre”.

Page 4: "Following maturation, the myofibre relies on re-establishment of connection with a motoneurone for full maturation into fiber type-specific myosins and other sarcomeric proteins". I would highlight a little more the impact of MN connection with MyHC switch since it's a key step in the transition from the regenerative to the mature fiber, and the nerve stimulation will mainly drive the phenotype of the fiber.

We have edited this paragraph to emphasize the necessity of the formation of the neuromuscular junction, starting Line 123:

“Newly formed myofibres express developmental myosin (Figure 3) (Ciciliot & Schiaffino, 2010), and rely on the re-establishment of connection with a motoneurone for full maturation. Without motoneurone input, the regenerating myofibres will remain small in size and will not be able to develop mature slow-type myosins and other sarcomeric proteins (Jerkovic et al., 1997; Ciciliot & Schiaffino, 2010). Once a neuromuscular connection is established, the fibre matures and can once again be recruited to contract and contribute to force production.”

Page 5: I would like to see the difference between fibrotic and scar tissue if there is any. This is a good question. When referring to fibrosis, we refer to the process of excess matrix deposition in general, both as a result of focal scar formation after tissue injury but also a general thickening of existing ECM, as affects many organs with age/disease. By using the term scar tissue, we mean the end-product of a “fibrotic” process, but confined to the site of tissue injury. We have added a bit to this paragraph (Line 218):

“It is interesting to note that while scar tissue (excess ECM at site of tissue injury) is a persistent and potentially permanent sign of prior injury, fibrosis (general thickening of existing ECM) does not develop with regeneration after contraction-induced necrosis (Mackey et al., 2016), again highlighting the fundamental difference in the pathology of, and repair capacity of the muscle in response to, strain injury versus contraction-induced myofibre necrosis. The persistence of scar tissue after strain injury suggests that, in place of proper tissue healing, the healing process results in failed reconstruction of the MTJ, with muscle fibres attaching to an adipose and disorganized matrix of scar tissue (Bayer et al., 2021) instead of direct attachment to the tendon. Indeed, the formation of such an inferior muscle-scar-tendon complex could explain the high recurrence rates of muscle strain injuries (Wangenstein et al., 2016), raising the question of how targeted interventions can shift this response towards more effective regenerative outcomes.”

The figures are really nice but not used enough. Fig1, the difference between NCAM and Pax7 staining is beautiful but not used by the authors. Also, for panel C, I would prefer to see the colors instead of black and white. Same for all the other figures.

Thank you for pointing this out. We have now referenced all our immunofluorescence figures more throughout the manuscript.

Regarding the greyscale colouring, we keep the single channel images in this format as the human eye sees best contrast in greyscale with poorer contrast between black and other colours (blue being the worst). The merged images are provided in colour which we hope will satisfy the reviewer.

Fi2, nice overview of the review but you wrote page 4 "muscle biopsies indicate that myofiber necrosis is rare". What about micro damages?

Good point, we see the contradiction. We have changed the figure legend for Figure 2 for clarification:

“B-C: Contraction-induced necrosis. B) Upon injury, such as extreme exercise involving repeated eccentric contractions, some myofibres will undergo full necrosis.”

We have also specified that necrosis is rare in the classic voluntary eccentric training studies (unlike NMES), and agree that micro-damage does likely occur. We have therefore changed the sentence starting on Line 137:

“However, analyses of muscle biopsies indicate that myofibre necrosis is rare in classical voluntary eccentric training contraction studies with signs instead of small localised myofibre damage (Yu et al., 2002; Crameri et al., 2007; Roman et al., 2021).”

REFERENCES

Grange S, Reurink G, Nguyen AQ, Riviera-Navarro C, Foschia C, Croisille P and Edouard P (2023). Location of Hamstring Injuries Based on Magnetic Resonance Imaging: A Systematic Review. *Sports health* 15, 111-123.

Karlsen A, Soendenbroe C, Malmgaard-Clausen NM, Wagener F, Moeller CE, Senhaji Z, Damberg K, Andersen JL, Schjerling P, Kjaer M and Mackey AL (2020). Preserved capacity for satellite cell proliferation, regeneration, and hypertrophy in the skeletal muscle of healthy elderly men. *FASEB J* 34, 6418-6436.

Lauritzen F, Paulsen G, Raastad T, Bergersen LH and Owe SG (2009). Gross ultrastructural changes and necrotic fiber segments in elbow flexor muscles after maximal voluntary eccentric action in humans. *J Appl Physiol* 107, 1923-1934.

Mackey AL, Rasmussen LK, Kadi F, Schjerling P, Helmark IC, Ponsot E, Aagaard P, Durigan JL and Kjaer M (2016). Activation of satellite cells and the regeneration of human skeletal muscle are expedited by ingestion of nonsteroidal anti-inflammatory medication. *FASEB J* 30, 2266-2281.

Murach KA, Mobley CB, Zdunek CJ, Frick KK, Jones SR, McCarthy JJ, Peterson CA and Dungan CM (2020). Muscle memory: myonuclear accretion, maintenance, morphology, and miRNA levels with training and detraining in adult mice. *J Cachexia Sarcopenia Muscle* 11, 1705-1722.

Dear Professor Mackey,

Re: JP-TR-2025-286507R1 "The repair capacity spectrum of human skeletal muscle injury from sports to surgical trauma settings" by Grith Højfeldt, Christian Hoegsbjerg, Arvind G. von Keudell, and Abigail Louise Mackey

We are pleased to tell you that your paper has been accepted for publication in The Journal of Physiology.

Authors should note that it is too late at this point to offer corrections prior to proofing. Major corrections at proof stage, such as changes to figures, will be referred to the Editors for approval before they can be incorporated. Only minor changes, such as to style and consistency, should be made at proof stage. Changes that need to be made after proof stage will usually require a formal correction notice.

Yours sincerely,

Paul Greenhaff
Senior Editor
The Journal of Physiology

P.S. - You can help your research get the attention it deserves! Check out Wiley's free Promotion Guide for best-practice recommendations for promoting your work at www.wileyauthors.com/eeo/guide. You can learn more about Wiley Editing Services which offers professional video, design, and writing services to create shareable video abstracts, infographics, conference posters, lay summaries, and research news stories for your research at www.wileyauthors.com/eeo/promotion.

IMPORTANT NOTICE ABOUT OPEN ACCESS: To assist authors whose funding agencies mandate public access to published research findings sooner than 12 months after publication, The Journal of Physiology allows authors to pay an Open Access (OA) fee to have their papers made freely available immediately on publication.

You can check if your funder or institution has a Wiley Open Access Account here: <https://authorservices.wiley.com/author-resources/Journal-Authors/licensing-and-open-access/open-access/author-compliance-tool.html>.

EDITOR COMMENTS

Reviewing Editor:

Thank you for your efforts in revising the manuscript so thoroughly. It is suitable for publication.

Senior Editor:

Thank you for the revised manuscript submission. I happy to say that both reviewers and the Reviewing Editor feel the authors have done a good job at revising the manuscript and believe it will be an influential review. Thank you for the additional efforts and choosing to publish your work in The Journal of Physiology.

REFEREE COMMENTS

Referee #1:

The authors have replied to my comments and for the most part performed adjustments to the text that I think have improved the quality of the manuscript. I have no further comments or concerns.

Referee #2:

Thank you for the great job the authors did with the manuscript. I enjoyed reading the responses as much as the edits to the manuscript.